# Patients with Kawasaki Disease Have Significantly Low Aerobic Metabolism Capacity and Peak Exercise Load Capacity during Adolescence

**DOI:** 10.3390/ijerph17228352

**Published:** 2020-11-11

**Authors:** Tsung-Hsun Yang, Yan-Yuh Lee, Lin-Yi Wang, Ta-Chih Chang, Ling-Sai Chang, Ho-Chang Kuo

**Affiliations:** 1Department of Physical Medicine and Rehabilitation, Kaohsiung Chang Gung Memorial Hospital and Chang Gung University College of Medicine, Kaohsiung 833, Taiwan; 8902077@cgmh.org.tw (T.-H.Y.); b9005009@cgmh.org.tw (Y.-Y.L.); s801121@cgmh.org.tw (L.-Y.W.); b333753@cgmh.org.tw (T.-C.C.); 2Department of Pediatrics, Kaohsiung Chang Gung Memorial Hospital and Chang Gung University College of Medicine, Kaohsiung 833, Taiwan; joycejohnsyoko@gmail.com; 3Kawasaki Disease Center, Kaohsiung Chang Gung Memorial Hospital, Kaohsiung 833, Taiwan

**Keywords:** Kawasaki disease, adolescent, cardiopulmonary functions, exercise testing, anaerobic threshold, motivation, self-efficacy

## Abstract

Introduction: Kawasaki disease (KD) is a childhood illness causing blood vessel inflammation. Children with KD have similar cardiopulmonary function to healthy children, but lower moderate-to-vigorous activity and exercise self-efficacy—possibly harming their cardiopulmonary function in adolescence. The purpose of this study is to investigate the cardiopulmonary function, exercise behaviors, exercise motivations, and self-efficacy of adolescents who once had KD. Methods: adolescents who once had KD and adolescents matched to the KD group in age and sex were enrolled. The cardiopulmonary exercise test was used to assess cardiopulmonary function. Weekly exercise behavior, exercise motivation, and self-efficacy were assessed with questionnaires. Results: this study recruited 50 and 30 participants, respectively, to the KD and control groups. The KD group had a lower ratio of VO_2_/kg at the anaerobic threshold and peak to the predicted VO_2_/kg at the peak (*p* = 0.021 and 0.043, respectively). No significant differences were found in questionnaire scores. The correlations of weekly exercise behavior scores with exercise motivation and self-efficacy scores were stronger in the KD group. Conclusions: adolescents with KD history had significantly lower aerobic metabolism capacity and peak exercise load capacity than controls. The correlations of amount of weekly exercise with exercise motivation and self-efficacy were stronger in the KD group.

## 1. Introduction

Kawasaki disease (KD) is an illness in children that causes inflammation of blood vessels throughout the body and primarily affects children younger than 5 years old. Approximately 1000 new cases of KD occur every year in Taiwan, placing Taiwan third worldwide, behind South Korea, for the number of individual cases. Japan has the highest incidence of 10,000–20,000 new cases each year. KD can cause inflammation of the coronary arteries, leading to coronary artery aneurysm (CAA), and is, therefore, the most common cause of acquired heart disease in children. Patients with coronary artery disease may need to take aspirin, a blood thinner, for the rest of their lives, as such disease may lead to heart failure and can thus be fatal. Furthermore, failure to treat child patients during the golden treatment period may result in severe heart disease sequelae.

Scholars have discovered that although children with KD have lower rates of myocardial perfusion while they are exercising, their cardiopulmonary function and exercise load capacities do not significantly differ from those of healthy children [1,2]. However, some studies have reported that children with KD have lower weekly moderate-to-vigorous activity (MVPA) than healthy children and lower exercise self-efficacy evaluations [3]. Adequate and regular exercise is critical to maintaining health; if children with KD have less MVPA in the long term than healthy children and are less willing to engage in athletic exercises, this may result in negative effects on their cardiopulmonary functions in adolescence. Furthermore, studies investigating KD have focused on children around the age of 10 years; no studies have yet been conducted regarding the cardiopulmonary functions of adolescents who had KD in childhood, nor have any studies analyzed the exercise behaviors, exercise motivations, and self-efficacy of this group. Therefore, the purpose of this study was to investigate the cardiopulmonary functions of adolescents with KD and to analyze their exercise behaviors, exercise motivations, and self-efficacy.

## 2. Methods

### 2.1. Experimental Design

The experiment was conducted through a cross-sectional, observational study.

### 2.2. Recruiting Participants

This study was conducted at a medical center in Southern Taiwan. The research team contacted patients listed in the medical center’s KD case management database, which had been established by the children’s rheumatology and immunology department. Contact was achieved through clinical recruitment, posters, and cold calling. The patients who agreed to participate in the study were assigned to the KD group. Age- and sex-matched healthy volunteers were recruited through posters advertising the study and assigned to the control group (hereafter referred to as the C group).

The inclusion criterion for the study was age 13–19 years. The exclusion criteria were (1) having had KD but the onset date could not be confirmed; (2) having a heart disease that can affect cardiopulmonary functions (such as moderate to severe valvular heart disease, heart failure, coronary artery diseases not caused by KD, severe arrhythmia, and ventricular hypertrophy); (3) having a pulmonary disease that can affect pulmonary functions (such as asthma or chronic obstructive pulmonary disease); and (4) other factors that would affect their results or that prevented the patient from accepting cardiopulmonary exercise tests or surveys.

### 2.3. Experimental Methods

First, the following information about the participants who met the inclusion and exclusion criteria and were willing to participate was recorded:Demographics: sex, age, height, weight, and body mass index (BMI).The recorded KD onset date for the KD group participants and number of years since the onset.In accordance with the KD group participants’ most recent echocardiography test results, Z score was calculated using the Taiwan Society of Pediatric Cardiology’s online calculator and the widest diameters of the left coronary artery (LCA) and right coronary artery (RCA) [4,5]. CAA was defined as Z score ≥2.5. The LCA or RCA Z score of the participants with CAA was recorded and ranked: Z score ≥2.5 and <5.0 indicates small aneurysms; ≥5.0 and <10.0 indicates large aneurysms; and ≥10.0 indicates giant aneurysms.Whether the KD group participants took aspirin as a blood thinner.After recording the participants’ personal information, the two groups underwent the cardiopulmonary exercise test (CPET) and survey assessments:Cardiopulmonary exercise test: the CPET was used to assess the participants’ cardiopulmonary functions and exercise load capacity performance. This test was conducted using the MasterScreen CPX (CareFusion Germany 234 GmbH, Hochberg, Germany). First, pulmonary function was evaluated in a resting state. Forced vital capacity (FVC), forced expiratory volume in 1 s (FEV1), and the FEV1/FVC ratio were recorded. Next, the participants were asked to put on masks connected to a gas analyzer and were connected to an electrocardiogram monitor and underwent treadmill exercise tests, according to the Bruce protocol [6]. The conditions for test termination were established based on the guidelines of the American College of Sports Medicine [7], which include the following: (1) the participants must subjectively feel that they have made the greatest effort and want the test to end so they can rest; (2) the participants felt any discomfort (including tightness in the chest, chest pain, shortness of breath, dizziness, or sore feet) and, therefore, unable or unwilling to continue the test; (3) the maximal oxygen consumption (VO_2_) plateaued within 2 min; (4) any myocardial ischemia should be observed (such as electrocardiogram showing the S-T band rising or falling) or arrhythmia occurred; and (5) other circumstances that may affect the safety of the CPET. Throughout the test, the participants’ blood pressure and heart rate were measured, and the gas analyzer measured the participants’ VO_2_, VCO_2_, and minute ventilation (VE) throughout the test. The participants’ respiratory exchange ratio (RER) was derived from the VCO_2_/VO_2_ ratio; RER ≥ 1.10 indicates that a patient has reached sufficient exercise intensity [7,8]. The VE/VO_2_ and VE/VCO_2_ values were employed to calculate the participants’ anaerobic threshold (AT)^9^. The AT is a significant increase in anaerobic glycolysis to provide energy when the oxygen supply in the circulatory system is insufficient to fulfill the current oxygen consumption levels at the exercise intensity [8]. The ratio of VO_2_/kg at the AT and the predicted VO_2_/kg at peak (hereafter referred to as AT%) were used to assess the aerobic metabolism capacity; a higher ratio indicates that a person’s aerobic metabolism and circulatory system oxygen supply capacity during exercise can withstand a high exercise load. VO_2_/kg at peak exercise intensity was used to reflect the participants’ peak exercise capacity [8]. Clinically, the ratio of this value to the predicted VO_2_/kg at peak (this ratio is hereafter referred to as Peak%) being 85% or higher is taken as the benchmark for “normal” test results [9]. The peak rate–pressure product (PRPP) can be used to reflect myocardial perfusion [7]. O_2_ pulse indicates oxygen intake and heart rate ratio and can be used to assess stroke volume [10]. The participants’ sex, age, height, and weight were input to the MasterScreen CPX, and the device then calculated their predicted FVC, FEV1, VO_2_/kg at peak, and O_2_ pulse, which were compared with the participants’ test data.The participants’ exercise behaviors were assessed using a Chinese version of the Godin Leisure-Time Exercise Questionnaire. This questionnaire was designed by G. Godin and R.J. Shephard [11,12] and assesses a person’s exercise behaviors using three simple questions. The observer asks the person to recollect how many times they had performed vigorous, moderate, and low-intensity exercise for more than 15 min on average each week in the preceding year. The score for weekly exercise behavior is calculated as follows: (9 × number of sessions of vigorous exercises) + (5 × number of sessions of moderate exercise) + (3 × number of sessions of low-intensity exercise).The participants’ exercise motivation and behavioral regulation were assessed using the Behavioral Regulation in Exercise Questionnaire 2nd edition (BREQ-2) in Chinese [13,14]. The BREQ-2 comprises 19 questions that determine a person’s reasons for exercising or not exercising as well as how they feel about exercise. These 19 questions have five dimensions related to exercise behavioral regulation: amotivation (e.g., “I don’t see why I should have to exercise”), external regulation (e.g., “I take part in exercise because my friends/family/partner say I should”), introjected regulation (e.g., “I feel guilty when I don’t exercise”), identified regulation (e.g., “I value the benefits of exercise”), and intrinsic regulation (e.g., “I find exercise a pleasurable activity”). Each question is scored between 0 and 4, with a higher score indicating stronger agreement with the statement. The questionnaire was graded and analyzed using two approaches. The first involved directly summing the individual scores for each dimension to determine the mental state in each dimension. The second involved multiplying the score for each dimension by the weight calculated from that dimension’s positive or negative effect on exercise motivation and degree of self-determination (the weights were −3 for amotivation, −2 for external regulation, −1 for introjected regulation, +2 for identified regulation, and +3 for intrinsic regulation); the weighted scores of all dimensions were summed to obtain the participants’ relative autonomy index (RAI), which reflected the participants’ autonomic exercise motivation.Self-efficacy for exercise was assessed using the Multidimensional Self-Efficacy for Exercise Scale in Chinese. This scale comprises nine statements that begin “How confident are you that you can…” and can be sorted into three efficacies relating to exercise: task efficacy (e.g., “…complete your exercise using proper technique”), coping efficacy (e.g., “…exercise when you lack energy”), and scheduling efficacy (e.g., “…consistently exercise three times per week”) [15]. The participants are asked to self-evaluate their degree of confidence regarding the circumstance described in the item on a scale of 0–10 (0 for *not confident at all* and 10 for *very confident*). This revealed the participants’ assessment of their own confidence in all dimensions.

### 2.4. Data Analysis

Statistical analysis of the data was conducted using MedCalc version 18 (MedCalc Software, Ostend, Belgium). The Kolmogorov–Smirnov test was employed to test the normality of the data distribution. The chi-squared test was used to compare the KD and C groups in terms of sex, distributions in age ranges, reasons for terminating the CPET, RER ≥ 1.10, and predicted number of participants with Peak% ≥ 85%. The CPET and survey results were verified using the independent-samples *t*-test. One way analysis of variance (ANOVA) was used to compare the CPET results of the KD group—participants with and without CAA—and C group, the CPET results between the KD group—taking and not taking aspirin—and C group, and CPET results within the KD group among participants with different degrees of CAA. Post hoc verification was conducted using the Scheffé test. Pearson’s correlation coefficients were calculated to determine the correlations between the Z score and CPET results, between the survey scores and CPET results, between the survey scores and body weight/BMI, and between several survey results. Correlation coefficients (CCs) were calculated, and the results were divided into four ranges: strong correlation (0.70–0.99), moderate correlation (0.40–0.69), weak correlation (0.10–0.39), and almost no correlation (0.01–0.09). The survey results of the five groups—group of participants with KD and CAA, group of participants with KD but without CAA, group of participants with KD who were taking aspirin, group of participants with KD who were not taking aspirin, and the C group—were compared using the Kruskal–Wallis test, and the post hoc test was conducted according to the method of Dunn [16,17]. In tests, *p* < 0.05 indicated a statistically significant difference.

### 2.5. Ethics

The study was conducted in accordance with the Declaration of Helsinki. The proposal for this study was reviewed and approved by the Chang Gung Medical Foundation Institutional Review Board (case no. 201601136B0C502). All participants were informed of the research process by the chief investigation prior to signing a consent form.

## 3. Results

### 3.1. Demographics

From January 2017 to December 2018, 88 people who met the inclusion criteria (56 and 32 in the KD and C groups, respectively) were recruited. Eight subjects refused to participate in the study (6 and 2 in the KD and C groups, respectively). Finally, 80 participants were enrolled in the study (50 and 30 in the KD and C groups, respectively).

Table 1 Details the sex, age, height, weight, and BMI of the participants as well as the age they received a diagnosis of KD and the number of years since the onset. The data for the two groups exhibited no significant differences.

Because physical development is rapid during adolescence, we classified all participants into three age ranges (13–15 years, 16–18 years, and 19 years) and analyzed the age distributions in both groups. In the KD group, 16 participants were aged 13–15 years (32%), 32 participants were aged 16–18 years (64%), and 2 participants were aged 19 years (4%); in the C group, 11 participants were aged 13–15 years (37%), 18 participants were aged 16–18 years (60%), and 1 participant was aged 19 years (3%). No significant difference existed between the age distributions in the two groups (*p* = 0.909).

### 3.2. Echocardiography of Coronary Arteries

Among the 50 participants in the KD group, 24 participants were revealed to have CAA in their echocardiography. The average peak Z score was 4.86 ± 1.53, and the average LCA Z score was 4.12 ± 1.12; the average RCA Z score was 4.20 ± 1.71. Among the 24 participants with CAA, 14 had small aneurysms, and 10 had large aneurysms; no participants had giant aneurysms.

### 3.3. Aspirin Use

Within the KD group, 18 participants regularly used aspirin as a blood thinner.

### 3.4. CPET Results

CPETs were successfully conducted for all the participants, and the electrocardiograms obtained in the testing process did not exhibit any abnormalities. Regarding the reasons the KD-group participants terminated their CPET, 27 participants ended their test due to soreness in their legs (54%) and 17 due to shortness of breath (34%). In four cases, the test was terminated because the participant’s VO_2_ plateaued (8%). In the remaining two cases, one participant’s chief complaint was foot pain, and the other participant asked to stop the test because of sweat in the eyes. In the C group, 14 participants stopped due to soreness in their legs (46.7%) and 10 from shortness of breath (33.3%). In six cases, the test was terminated because the participant’s VO_2_ plateaued (20%). Comparison of the reasons for test termination in the two groups yielded no significant difference (*p* = 0.534). Whether the participants achieved sufficient exercise intensity (RER ≥ 1.10) was analyzed; 42 participants in the KD group achieved sufficient exercise intensity (84%), and 8 participants did not; 7 of those 8 participants stopped the test due to soreness in their legs, and the 8th stopped the test due to foot pain. In the C group, 27 participants achieved sufficient exercise intensity (90%), and 3 did not; 2 of the 3 stopped the test due to soreness in their legs and the other due to shortness of breath. No significant intergroup difference was determined regarding whether sufficient exercise intensity was achieved (*p* = 0.453).

Table 2 compares the CPET results of the two groups. The KD group had lower AT% and Peak% than the C group. Furthermore, although the result was nonsignificant, the ratio of KD participants whose Peak% exceeded 85% was lower than in the C group (16/50 vs. 16/30, *p* = 0.061). The other data did not exhibit any significant intergroup differences.

Comparison of the CPET results between the KD + CAA group, KD + no CAA group, and C group was performed. One-way ANOVA of AT% determined significant differences between the three groups (*p* = 0.036), and using the Scheffé test in post hoc pairwise comparisons indicated no significant differences between the three groups. No significant intergroup differences were observed in other data (Appendix A).

The participants without CAA (Z score < 2.5), with small aneurysms (Z score ≥2.5 to <5.0), and with large aneurysms (Z score ≥5.0 to <10.0) within the KD group were compared, and no significant differences were discovered between these groups for VO_2_/kg at AT, AT%, VO_2_/kg at peak, Peak%, peak RER, and PRPP (*p* = 0.527, 0.930, 0.686, 0.892, 0.482, and 0.874, respectively). Correlation analysis of Z scores and CPET results for the participants with CAA revealed that although the RCA Z score and CPET results were not significantly correlated, there was a trend showing their negative correlation with VO_2_/kg at peak (CC = −0.377, *p* = 0.070). Peak Z score was not significantly correlated with LCA Z score or CPET results.

Comparison of the CPET results between the KD group participants who regularly took aspirin, KD group participants who did not take aspirin, and C group participants was also performed. One-way ANOVA of AT% revealed significant intergroup differences (*p* = 0.038), and using the Scheffé test in post hoc pairwise comparisons revealed no significant differences between the three groups. No significant differences were observed in other data between the three groups (Appendix A).

### 3.5. Survey Results

Table 3 details the two groups’ survey scores. No significant differences were discovered in item scores or total scores for the Godin Leisure-Time Exercise Questionnaire, RAI for individual items and weighted calculations in the BREQ-2, or Multidimensional Self-Efficacy for Exercise Scale. One study found that patients with KD have lower MVPA than healthy people^3^. The Godin Leisure-Time Exercise Questionnaire responses of the KD and C groups were compared, especially the number of times the participants engaged in moderate and vigorous exercise and the total score; however, no significant differences were discovered between the two groups in the number of times they engaged in MVPA (KD group 3.60 ± 4.62 vs. C group 4.33 ± 4.20, *p* = 0.479) or total score (KD group 28.22 ± 28.75 vs. C group 31.53 ± 33.15, *p* = 0.639).

The KD group was divided into four groups based on whether the participant had CAA and whether they regularly took aspirin; they formed the without CAA and not taking aspirin (*n* = 19), with CAA and not taking aspirin (*n* = 13), without CAA and taking aspirin (*n* = 8), and with CAA and taking aspirin (*n* = 10) groups. These four groups were then compared with the C group; however, these comparisons revealed no significant differences.

Table 4 presents the correlation analysis results for the Godin Leisure-Time Exercise Questionnaire, BREQ-2, and Multidimensional Self-Efficacy for Exercise Scale scores. For the KD group, their Godin Leisure-Time Exercise Questionnaire score was weakly negatively correlated with amotivation score (CC = −0.317, *p* = 0.025) and external regulation (CC = −0.353, *p* = 0.012), weakly positively correlated with identified regulation (CC = 0.387, *p* = 0.006) and intrinsic regulation (CC = 0.335, *p* = 0.018), and moderately positively correlated with RAI score (CC = 0.436, *p* = 0.002), weakly positively correlated with corresponding efficacies (CC = 0.376, *p* = 0.007) and moderately positively correlated with planning efficacy (CC = 0.486, *p* < 0.001) and total score in the Multidimensional Self-Efficacy for Exercise Scale (CC = 0.442, *p* = 0.001). For the C group, a moderately positive correlation was discovered between the Godin Leisure-Time Exercise Questionnaire score and intrinsic regulation in BREQ-2 (CC = 0.477, *p* = 0.008), but no significant correlations between other BREQ-2 items and multidimensional self-efficacy for exercise items were found.

Correlation analysis of the survey and CPET results revealed a weak negative correlation between Peak% and BREQ-2 external regulation score in the KD group (CC = −0.354, *p =* 0.012) and a moderate positive correlation between Peak% and BREQ-2 external regulation score in the C group (CC = 0.510, *p* = 0.004). Neither the Godin Leisure-Time Exercise Questionnaire scores nor Multidimensional Self-Efficacy for Exercise Scale scores were significantly correlated with the CPET results. Correlations between body weight, BMI, and the survey results were analyzed, but no significant correlation was found.

## 4. Discussion

This is the first study to investigate the cardiopulmonary functions and exercise load capacity of adolescents who once had KD. Compared with typical adolescents, adolescents who had once had KD were discovered to have lower aerobic metabolism capacity and peak exercise load capacity. This result supports our hypothesis that children with KD have less MVPA than do their healthy peers, which may negatively affect their cardiopulmonary function in adolescence in the long term. In a study conducting treadmill-based CPETs of individuals who had once had KD, Tuan et al. discovered that their aerobic metabolism capacity and peak exercise load capacity were non-significantly different from those of the control group [2]. The age range in that study was 5–18 years, and the average participant age was 12.27 ± 3.76 years; only 12 of the 63 participants had received a diagnosis of KD within the 5 years prior to the study (the study did not report the average number of years since KD onset). The age range for the KD group in the present study was 13–19 years, and the average age was 15.98 ± 1.85 years; the average time since onset of KD was 14.08 ± 2.85 years. Compared with the study of Tuan et al., which included children who had not yet begun adolescence, the participants in the present study were all teenagers, and the KD diagnosis had been made longer ago. This discrepancy between the two studies indicates that the difference in daily MVPA [3] may not have a strong effect in childhood, but may lead to lower aerobic capacity and peak exercise capacity in adolescents with KD compared with their peers. Although significant differences were not discovered in the weekly exercise load of the two groups in this study, the data were obtained through the participants’ self-evaluation by using the Godin questionnaire, and their responses may not have reflected their actual exercise load; additionally, their responses may not have reflected their activity that was not exercise behaviors. With the continuing advancement of sports-related wearable technology, future studies may consider testing participants’ daily activity levels by using wearable technology to determine whether the daily activity levels of adolescents who had once had KD and typical adolescents are different and to investigate the effects of previous KD on cardiopulmonary functions. Furthermore, vasculitis and CAA caused by KD may lessen as KD condition improves, but Iemura et al. discovered that 10 years after KD treatment, patients’ coronary arteries still had thickened inner walls, significant contractions, and poor diastolic function [18]; consequently, the effects of KD on the circulatory system years later cannot be discounted, and whether this factor affects the cardiopulmonary functions of patients with KD who are entering adolescence should be investigated.

The one-way ANOVA of the CPET results of KD patients with or without CAA revealed significant differences among the KD + CAA group, KD + no CAA group, and C group. However, post-hoc pairwise comparisons revealed no significant differences among the three groups. Similar findings were obtained from the analysis of CPET results of KD patients with or without aspirin use. This indicated that CAA or aspirin was not an indicator of CPET results, and the differences in AT% were mainly between the KD group and C group.

Although the results of this study indicate that adolescents who had once had KD had lower aerobic metabolism capacity and peak exercise load capacity, this does not mean they are unable to engage in exercise normally. The World Health Organization (WHO), recommends 75 min of vigorous aerobic exercise and 150 min of moderate aerobic exercise per week [19]. The American College of Sports Medicine defines vigorous exercise as 21.0–30.5 mL/min of VO_2_/kg^8^. In this study, the average peak exercise load in the KD group in the CPET was 33.63 ± 6.43 mL/min/kg, which exceeds the requirements for most vigorous exercises, indicating that the KD group may engage in normal daily exercise safely. Notably, the percentage of participants whose Peak% exceeded 85% was low in both the KD group (32%) and the C group (53%), suggesting that adolescents in both groups engage in insufficient physical activity. According to the latest survey by the WHO, the global prevalence of insufficient physical activity in adolescents is 81.0%. In Taiwan, where this study was conducted, 84.4% of adolescents engage in insufficient physical activity, a worse rate than the global average [20]. Thus, it is crucial to emphasize the importance of a physically active lifestyle and encourage people to engage in regular exercise.

In addition to investigating cardiopulmonary functions, this was the first study to evaluate the exercise behaviors, exercise motivations, and self-efficacy of adolescents who had once had KD. This study discovered that compared with peers in the same age group, adolescents who had once had KD did not have a significantly different self-assessed amount of weekly exercise, exercise motivation of any type, or task efficacy, coping efficacy, or scheduling efficacy. Moreover, division of the KD group on the basis of whether the participants had CAA or regularly used blood thinners and subsequent analysis revealed no significant differences in self-assessed amount of weekly exercise, exercise motivation, and self-efficacy, indicating that CAA and use of blood thinners do not appear to affect the amount of exercise performed by adolescents who had once had KD or their motivations and confidence regarding exercise. Correlation analysis demonstrated that in the KD group, the participants’ amount of weekly exercise was moderately positively correlated with RAI—which reflects overall exercise motivation—and overall self-efficacy score; conversely, in the control group, the amount of weekly exercise was moderately positively correlated with only intrinsic regulation among all the types of exercise motivation; no significant correlations were discovered with other exercise motivations or self-efficacies. This indicates that the exercise behaviors of adolescents who had once had KD were more likely to be affected by exercise motivation and self-efficacy, compared with the behaviors of typical adolescents. Therefore, how the exercise motivation and confidence of this group, and consequently also their health and cardiopulmonary fitness, can be improved through approaches such as health education campaigns and an upgrade to exercise environments to improve is worthy of research. The positive effects of improving the exercise motivation and confidence of this population may be even more significant than for typical adolescents.

This study had several limitations. First, the sample was relatively small, and second, the participants were recruited from one medical center and thus their circumstances may not reflect the circumstances of all adolescents who had once had KD. Third, in this study, Z scores were calculated using the online calculator provided by the Taiwan Society of Pediatric Cardiology, because this is the only algorithm constructed to be applicable to Taiwanese people. However, the algorithm was developed using data only from subjects aged 6 years and under [5]; therefore, there may be errors in the calculations of Z scores for adolescents. Fourth, exercise behaviors were collected using the self-evaluative Godin Leisure-Time Exercise Questionnaire; self-evaluations may be subject to memory errors and cannot reflect all activities in daily life. Future studies should use a larger sample and recruit participants from multiple locations, even from overseas; additionally, a Z score calculation formula that is appropriate to the participants of interest should be employed or developed. Moreover, wearable technology could be employed to accurately calculate participants’ daily activity levels. These additional measures would clarify whether the cardiopulmonary functions and exercise loads of adolescents who had once had KD are impaired and the exercise behavior and activity level differences between this population and typical adolescents.

## 5. Conclusions

Compared with control adolescents, adolescents who had KD history had lower aerobic metabolism capacity and peak exercise load capacity, which should not affect their ability to engage in daily leisure exercise. CAA and regular use of blood-thinning medication did not significantly affect aerobic metabolism capacity and peak exercise load capacity, nor did it have significant effects on daily exercise, exercise motivation, and self-efficacy. Furthermore, for the adolescents who once had KD, significant positive correlations were discovered between amount of weekly exercise and exercise motivation and between amount of weekly exercise and self-efficacy; these correlations were stronger than in peers of the same age.

## Figures and Tables

**Table 1 ijerph-17-08352-t001:** General characteristics of the participants in each group.

	KD Group (*n* = 50)	Control Group (*n* = 30)	*p*
Sex (male/female)	33/17	15/15	0.160
Age (year)	15.98 ± 1.85	15.90 ± 1.83	0.971
Height (cm)	165.01 ± 8.67	164.88 ± 9.30	0.651
Weight (kg)	61.99 ± 15.58	57.87 ± 13.04	0.308
BMI (kg/m^2^)	22.60 ± 4.46	21.16 ± 3.53	0.136
Age of KD diagnosis (year)	2.58 ± 1.99		
Years since KD diagnosis (year)	14.08 ± 2.85		

Values are expressed as mean ± standard deviation. The chi-square and Student’s *t*-test tests were used to compare differences between the two groups, KD: Kawasaki disease; BMI: Body mass index.

**Table 2 ijerph-17-08352-t002:** Results of CPET in each group.

	KD Group (*n* = 50)	Control Group (*n* = 30)	*p*
FVC (L)	3.56 ± 0.82	3.34 ± 0.74	0.234
FVC% (%)	90.84 ± 11.68	88.22 ± 11.95	0.338
FEV1 (L)	3.21 ± 0.71	3.00 ± 0.66	0.187
FEV1% (%)	97.51 ± 11.96	93.96 ± 14.19	0.235
FEV1/FVC (%)	90.53 ± 5.56	89.96 ± 8.42	0.743
VO_2_/kg at AT (mL/min/kg)	24.56 ± 5.29	26.27 ± 7.53	0.279
AT% (%)	57.19 ± 11.38	65.36 ± 16.45	0.021 *
VO_2_/kg at peak (mL/min/kg)	33.63 ± 6.43	35.20 ± 9.32	0.419
Peak% (%)	78.90 ± 16.38	86.83 ± 17.15	0.043 *
Peak% exceeded 85% (Yes/No)	16/34	16/14	0.061
Peak O_2_ pulse (mL/beat)	11.17 ± 2.87	11.05 ± 3.46	0.862
O_2_ pulse% (%)	90.06 ± 15.47	92.27 ± 22.28	0.635
RER at peak	1.18 ± 0.09	1.20 ± 0.12	0.426
PRPP	31,187.42 ± 4114.64	31,261.03 ± 4279.49	0.940

Values are expressed as mean ± standard deviation. The Independent samples *t*-test was used to compare differences between the two groups. CPET: Cardiopulmonary exercise test; KD: Kawasaki disease; FVC: forced vital capacity; FVC%: percentage of FVC compared with predicted FVC; FEV1: forced expiratory volume in one second; FEV1%: percentage of FEV1 compared with predicted FEV1; VO_2_: oxygen uptake; AT: anaerobic threshold; AT%: percentage of VO_2_/kg at AT compared with predicted peak VO_2_/kg; Peak%: percentage of VO_2_/kg at peak compared with predicted peak VO_2_/kg; O_2_ pulse%: percentage of peak O_2_ pulse compared with predicted peak O_2_ pulse; RER: Respiratory exchange ratio; PRPP: peak rate-pressure product, * *p* < 0.05.

**Table 3 ijerph-17-08352-t003:** Results of Godin leisure-time exercise questionnaire, BREQ-2, and multidimensional self-efficacy for exercise scale in each group.

	KD Group (*n* = 50)	Control Group (*n* = 30)	*p*
Godin leisure-time exercise questionnaire (score)	33.32 ± 32.22	38.63 ± 31.72	0.475
BREQ-2			
Amotivation (score)	2.36 ± 2.96	2.87 ± 3.63	0.498
External regulation (score)	4.14 ± 3.55	4.23 ± 3.57	0.910
Introjected regulation (score)	3.44 ± 3.16	4.03 ± 3.02	0.411
Identified regulation (score)	8.42 ± 4.12	9.10 ± 2.93	0.431
Intrinsic regulation (score)	11.82 ± 3.85	11.77 ± 3.22	0.949
RAI (score)	33.74 ± 28.74	32.80 ± 23.99	0.881
Multidimensional self-efficacy for exercise scale			
Task efficacy (score)	19.60 ± 7.04	20.67 ± 5.26	0.475
Coping efficacy (score)	11.18 ± 8.06	10.47 ± 5.53	0.641
Scheduling efficacy (score)	18.16 ± 8.40	19.40 ± 6.57	0.492
Total score (score)	48.94 ± 20.34	50.53 ± 14.22	0.682

Values are expressed as mean ± standard deviation, the Independent samples *t*-test was used to compare differences between the two groups. BREQ-2: The Behavioral Regulation in Exercise Questionnaire, 2nd edition; KD: Kawasaki disease; RAI: relative autonomy index.

**Table 4 ijerph-17-08352-t004:** Correlation coefficients between Godin leisure-time exercise questionnaire, and BREQ-2, multidimensional self-efficacy for exercise scale in each group.

	KD Group (*n* = 50)		Control Group (*n* = 30)	
	Godin Leisure-Time Exercise Questionnaire (Score)	*p*	Godin Leisure-Time Exercise Questionnaire (Score)	*p*
BREQ-2				
Amotivation (score)	−0.317	0.025 *	−0.149	0.433
External regulation (score)	−0.353	0.012 *	−0.204	0.281
Introjected regulation (score)	0.225	0.116	0.103	0.588
Identified regulation (score)	0.387	0.006 *	−0.007	0.971
Intrinsic regulation (score)	0.335	0.018 *	0.477	0.008 *
RAI (score)	0.436	0.002 *	0.276	0.140
Multidimensional self-efficacy for exercise scale				
Task efficacy (score)	0.265	0.062	0.053	0.783
Coping efficacy (score)	0.376	0.007 *	0.218	0.247
Scheduling efficacy (score)	0.486	<0.001 *	0.147	0.438
Total score (score)	0.442	0.001 *	0.172	0.363

The Pearson’s correlation coefficient was used to assess the correlation coefficients. BREQ-2: the Behavioral Regulation in Exercise Questionnaire, 2nd edition; KD: Kawasaki disease; RAI: relative autonomy index, * *p* < 0.05.

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
