# Peer review of "Patients with Kawasaki Disease Have Significantly Low Aerobic Metabolism Capacity and Peak Exercise Load Capacity during Adolescence"

_ijerph, 2020, doi:10.3390/ijerph17228352_

Round 1
Reviewer 1 Report
The authors have made the appropriate additions.
We may wish the authors in the future to use Figures rather than of Тables.
Author Response
Thank you for your opinion.
Reviewer 2 Report
The paper from Tsung-Hsun Yang et al, has the purpose to investigate the cardiopulmonary function, exercise behaviors, exercise motivations, and self-efficacy of adolescents who once had KD.
There are few publications in this field as confirmed by the few references in the article, so it's really important the contribute that this paper give.
The paper is potentially of interest and suitable for IJERPH
Moreover, the study was conducted with scientific rigor and following all the criteria that allow to obtain a good result about this topic.
Nevertheless, In my opinion the manuscript needs a minor revision to better highlight the results and topics.
1)the author should clarify, since they decided to create just one group in which put all together adolescent (13-15), young adult (16-18) and adult (19) years, if they try to analyze the results considering these difference.
2)Did you performed the correlation analysis between the results and the BMI?
3)In the table 4 there are the correlation coefficients between Godin leisure-time exercise questionnaire, and BREQ-2, multidimensional self-efficacy for exercise scale in each group.
Since many of the results are statistical significant should be better reorganized the dates because are not clear and the column results staggered and don't allow to highlight the results.
4)In table A2 concern the results of CPET for participants with or without aspirin use, I didn't find any discussion about the result AT% between the two groups.
Author Response
Please see the attachment.

This manuscript is a resubmission of an earlier submission. The following is a list of the peer review reports and author responses from that submission.
Round 1
Reviewer 1 Report
The work is devoted to the problem of children's health. In particular Kawasaki disease in children that causes inflammation of blood vessels throughout younger than 5 years old. The purpose of this study was to investigate the cardiopulmonary functions of adolescents with Kawasaki disease and to analyze their physical exercise behaviors, exercise motivations, and self-efficacy.
- The conclusion is consistent with the purpose of this study
- The test assessing cardiopulmonary functions is described in detail
- Logically matched groups of subjects
- Used a wide range of statistical analysis to process the data obtained
- The list of references corresponds to the content of the manuscript.
Author Response
Thank you for your opinion.
Reviewer 2 Report
The manuscript entitled ‘Kawasaki disease patients had significantly lower aerobic metabolism capacity and peak exercise load capacity during adolescents’ by Yang et al., is a study of adolescents aged 13 to 19 years who had Kawasaki disease. The authors studied several cardiopulmonary parameters and also studied participants exercise motivation with previously established questionnaires.
Comments
- The study has small number of patients and the authors have discussed that issue. Can the authors comment or further discuss on any cardiopulmonary parameter differences associated with gender?
- The Reference number 2 that the authors have cited (Tual et al., 2016) has studied patients in the age group 5 – 18 years. The authors should discuss the results with that work comparatively.
- There is already existing literature and data on the long-term effect of Kawasaki patients. And therefore, the contribution of the currently study is unclear.
- The authors seem to have compared the CPET results of patients with aneurysm. However, the results (Table A1) is missing from the text. Similarity, Table A2 is missing.
- Manuscript has few grammatical errors. It should be checked by a native English speaker.
Author Response
Response to Reviewer 2 Comments
Point 1: The study has small number of patients and the authors have discussed that issue. Can the authors comment or further discuss on any cardiopulmonary parameter differences associated with gender?
Response 1: The main outcomes of the exercise test conducted in this study were AT% and Peak%, which represent the ratios of VO2/kg at the anaerobic threshold (AT) and at peak workload, respectively, to the predicted peak VO2/kg. One of the factors used for predicting VO2/kg at peak was gender (the others were age, height, and weight). Therefore, gender differences did not affect AT% and Peak% because they were already corrected for in the calculation.
Point 2: The Reference number 2 that the authors have cited (Tual et al., 2016) has studied patients in the age group 5 – 18 years. The authors should discuss the results with that work comparatively.
Response 2: Thank you for your opinion. The manuscript has been revised, and the related discuss has been included in the Discussion section, on line 300-312.
Point 3: There is already existing literature and data on the long-term effect of Kawasaki patients. And therefore, the contribution of the currently study is unclear.
Response 3: Most studies on the long-term effect of Kawasaki disease (KD) have discussed the status of patients’ coronary arteries and their heart function, but to our best knowledge, no articles have reported on the long-term effect of KD in adolescence on cardiopulmonary performance. Our study is the first to investigate this subject, and it revealed a relative deficit in aerobic metabolism capacity and peak exercise load capacity in patients who had once had KD compared with their healthy peers.
Point 4: The authors seem to have compared the CPET results of patients with aneurysm. However, the results (Table A1) is missing from the text. Similarity, Table A2 is missing.
Response 4: Table A1 and A2 were renamed as Table S1 and S2, and included in supplementary materials.
Point 5: Manuscript has few grammatical errors. It should be checked by a native English speaker.
Response 5: Thank you for your opinion. The manuscript had been checked and revised by Wallace Academic Editing.
Reviewer 3 Report
Had lower aerobic metabolism capacity (unnecessary to mention as Human cells can only perform that mechanism)
Notably, the percentage of participants whose Peak% exceeded 85% was 320 low in both the KD group (32%) and the C group (53%), suggesting that adolescents in both groups 321 engage in insufficient physical activity: What is the significance of the current study (no difference between KD patients who took blood thinners and C group etc)? This seems more like a pilot study. Moreover, the study is a self assessment test of participants and lacks scientific soundness.
What are the possible mechanisms involved for lower exercise capacity (scientifically)?
Author Response
Response to Reviewer 3 Comments
Point 1: Had lower aerobic metabolism capacity (unnecessary to mention as Human cells can only perform that mechanism)
Response 1: In this article, aerobic metabolism capacity indicates the exercise intensity at the anaerobic threshold (AT). If the exercise intensity exceeds the AT, then the proportion of anaerobic metabolism dramatically increases because aerobic metabolism cannot complete the high workload.
Point 2: Notably, the percentage of participants whose Peak% exceeded 85% was low in both the KD group (32%) and the C group (53%), suggesting that adolescents in both groups engage in insufficient physical activity: What is the significance of the current study (no difference between KD patients who took blood thinners and C group etc)? This seems more like a pilot study. Moreover, the study is a self-assessment test of participants and lacks scientific soundness.
Response 2: Thank you for your opinion. Clinically, a Peak%, which is the ratio of VO2/kg to the predicted peak VO2/kg, of 85% or higher is taken as the benchmark for a “normal” test result. The present study showed that even in the control group, only 53% of the participants reached this “normal” standard. Our hypothesis is that this may have resulted from insufficient physical activity because normal results were obtained in pulmonary function tests and for AT% in almost every participant with low peak exercise capacity. (Clinically an AT% of 40% or higher is taken as benchmark for a “normal” test result. https://www.ahajournals.org/doi/10.1161/01.CIR.0000136811.45524.2F) This suggests that the major cause of low peak exercise capacity was deconditioning, which results in relatively low musculoskeletal and cardiopulmonary endurance. However, we require further scientific evidence to prove this hypothesis (for example, a wearable device could be used to record physical activity).
Point 3: What are the possible mechanisms involved for lower exercise capacity (scientifically)?
Response 3: A person’s exercise capacity is the sum of multiple factors, including musculoskeletal condition, circulatory function, ventilatory function, and motivation for performing exercise. Several conditions observed and values recorded during exercise tests could help us determine the cause of low exercise capacity, including the reason for termination, respiratory exchange ratio (RER), pulmonary function test, AT%, O2 pulse%, and so forth. The mechanisms involved in low exercise capacity are different in each case and must be analyzed individually.
Round 2
Reviewer 2 Report
The manuscript by Yang et al. has shown a minor improvement in discussion section. The authors mention that the manuscript has been checked by Wallace Academic Editing. This is unclear as the changes are not visible in the revised version. E.g. the title needs rephrasing (Kawasaki disease patients HAVE significantly lower aerobic metabolism capacity and peak exercise load capacity during ADOLESCENCE). The tenses have not been correctly used in many sentences.
Line 381, ‘Furthermore,…’sentence has incorrect use of had twice.
Also, in the Table 4, the labels of the columns are not properly aligned, and n is missing for KD and control group in this table compared to other tables.
Reviewer 3 Report
With respect to Question2: If the control group was incapable of reaching the required 85% or higher of VO2/kg due to insufficient collection of data (insufficient exercise etc), how credible is the study and its comparison to other groups?
With respect to Question 3: Separate cohorts could be studied which could be represented as data specific for the scientific credibility of the current study. Without which, the current study is quasi to a pilot study.